# Thousands of Women’s Lives Depend on the Improvement of Poland’s Cervical Cancer Screening and Prevention Education as Well as Better Networking Strategies Amongst Cervical Cancer Facilities

**DOI:** 10.3390/diagnostics12081807

**Published:** 2022-07-26

**Authors:** Marcin Śniadecki, Patryk Poniewierza, Paulina Jaworek, Ada Szymańczyk, Gorm Andersson, Maria Stasiak, Michał Brzeziński, Małgorzata Bońkowska, Magdalena Krajewska, Joanna Konarzewska, Dagmara Klasa-Mazurkiewicz, Paweł Guzik, Dariusz Grzegorz Wydra

**Affiliations:** 1Department of Obstetrics and Gynecology, Gynecological Oncology and Gynecological Endocrinology, Medical University of Gdańsk, 80-210 Gdańsk, Poland; pjaworek@gumed.edu.pl (P.J.); ada.anastazja@gumed.edu.pl (A.S.); gorm_andersson@gumed.edu.pl (G.A.); mariastasiak@gumed.edu.pl (M.S.); magdalena_krajewska@gumed.edu.pl (M.K.); dklasa@gumed.edu.pl (D.K.-M.); dgwydra@gumed.edu.pl (D.G.W.); 2Medical Department, Medicover Sp. z o.o., 00-807 Warsaw, Poland; pponiewierza@wsiiz.pl; 3Department of Gynecologic Oncology, PCK Marine Hospital in Gdynia, 81-519 Gdynia, Poland; m.brzezinski@gumed.edu.pl; 4Emergency Department, University Clinical Center in Gdańsk, 80-952 Gdańsk, Poland; bonkowska.malgorzata1@gmail.com; 5Department of Radiology, Medical University of Gdańsk, 80-210 Gdańsk, Poland; mijo@gumed.edu.pl; 6Clinical Department of Gynecology and Obstetrics, City Hospital Rzeszów, 35-241 Rzeszów, Poland; pawelguzik@gmail.com

**Keywords:** cervical cancer, screening program, education, progression

## Abstract

Proper targeted cancer prophylaxis reduces the incidence of cancer in all forms; this includes cancers with significant progression potential and poor prognosis. Based on the assumption that one of the risk factors of cervical cancer is the avoidance of screening tests, we analyzed the current scenario of cervical cancer (CC) screening and recommendations in Poland (country with a well-off socioeconomic status). Based on the comprehensive literature review concerning documents of guidelines and recommendations of various bodies, including national ones, data on the implementation of CC screening in Poland, and different models for medium-to-high-income countries, we proposed how the CC screening strategy could be improved. Finally, the new strategy was further developed for those who are prone to not being screened. The proposal on how to improve the Polish CC screening program is the following: refinement of the public education on CC risk factors, popularization of CC screening incentives amongst the public, and improvement of networking strategies between CC screening facilities (“cervical screening clinical”), allowing screenings to be more efficient and rapid. We believe that, to enhance the future quality of life of those with rapid CC progression by catching the disease preemptively and limiting the sequelae of the disease, we have to improve education and access to medical services.

## 1. Introduction

Cervical cancer (CC) is the most prevalent gynecological malignancy worldwide [1]. Three thousand women are newly diagnosed with CC in Poland each year [2]. Despite the fact that the registered absolute number of CC cases in the country counting 38 million people is not very high, it should be noted that the five-year overall survival (OS) rate for CC is only about 54%, and each individual CC patient through her disease has a negative psychological and economic impact on her relatives and, thus, for the whole society [3]. It is obvious that this mainly happens when the disease has progressed to an incurable stage.

Moreover, according to the International Agency for Research on Cancer (IARC), the age-standardized mortality rate (deaths from CC per 100,000 women/year) for Poland is 4.9, and the European average is 3.1. This value is comparable to Eastern European countries, with their widely considered worse access to medical care. According to the report, Poland is the seventh country in Europe with the highest mortality rate for CC [4].

Initially, it seemed that the breakthrough moment in the fight against CC was the implementation of vaccinations against the high-risk human papilloma virus (hrHPV), which (as a group of viruses) is responsible for about 99.7% of CC cases [5]. The effectiveness of the vaccines against hrHPV is relatively high, especially if they are administered before sexual activity is initiated [6].

Unfortunately, hardly anyone predicted that this extremely effective tool in terms of preventing CC would meet with such a lot of reluctance among society to use it. In Poland, it is estimated that the vaccination coverage accounts only about 7.5–10% [7]. This value is one of the lowest among European countries—United Kingdom, Spain, Portugal, Norway, Sweden, Iceland, and the Belgian Flanders exceeded the threshold of 70% (Figure 1) [8,9]. The estimated level of full-course vaccination of society against HPV in various countries in the European Union (EU) is shown in Figure 2. Poland has one of the lowest levels in the EU.

The Human Development Index (HDI) in 2019 for Poland was 0.88, which places Poland between developed and developing European countries [10]. According to our preliminary report, the level of participation in the prevention program is insufficient. The main aim of the current study was to determine how to improve the implemented CC screening program. Whilst focusing on this aim, special attention was paid to those who had not been attending CC screening tests and, thus, currently have an increased risk of CC with further progression [11].

## 2. Materials and Methods

Firstly, the current study is based on our preliminary report, which analyzed the financial and epidemiological data from the period of 2011–2017, provided by the Polish National Health Fund and the Polish National Cancer Registry, on the prevention and treatment of CC in Poland [11]. Secondly, based on the comprehensive literature review, concerning documents of guidelines and recommendations of various bodies, including national ones, of data on the implementation of CC screening in Poland, different models for medium-to-high-income countries, public audience resources (social stakeholder group), and reports on non-screened groups, we proposed how the CC screening strategy could be improved. Finally, the new strategy was further developed for those who are prone to not being screened.

## 3. Results

### 3.1. State of Art

#### 3.1.1. Primary Prevention

Primary prevention pertains to the successful HPV vaccination program directed via the government guidelines. In Poland, there are three types of vaccines against HPV available. The first is the bivalent vaccine Cervarix (HPV-2), which is targeted against 16 and 18 HPV variants. Next is the quadrivalent HPV vaccine Gardasil (HPV-4) targeted against 6, 11, 16, and 18 HPV variants, and the final nine-valent vaccine Gardasil 9 (HPV-9) is targeted against 6, 11, 16, 18, 31, 33, 45, 52, and 58 HPV variants.

These vaccines are promoted to both genders. The scheme of HPV vaccine dosing in Poland includes children between the ages of 9 and 14 years old to receive two doses of the HPV-2 vaccine, the second dose is to be given 5–13 months apart from the first dose. If the child receives the second dose earlier than 5 months, a third dose of the HPV-2 vaccine must be administered. The dosing scheme of the HPV-4 and HPV-9 vaccines for children between the ages of 9 and 14 years old is exactly the same as for the scheme of the HPV-2 vaccine.

Regarding individuals aged 15 years old or older, three doses of the HPV-2 vaccine are required. After the first dose, 1 month must pass for the injection of the second dose, and then 6 months must pass to receive the final third dose. Meanwhile, for individuals 15 years old or older receiving the HPV-4 or HPV-9 vaccines, the time spacing between the doses is different. After the first dose, the second dose is given after 2 months, then the third dose is administered after 6 months.

In Poland, the HPV-2 vaccine effectiveness is based on the results in women between the ages of 15 and 25 years old and has a proven immunogenicity in females ranging from 9 to 25 years old. The HPV-4 effectiveness is based on the results from women ranging between 16 and 26 years old and has a proven immunogenicity in females ranging between 9 and 15 years old.

The Polish government states that 50–80% of sexually active men and women will be infected with HPV, half of those individuals being between the ages of 15 and 25 years old. The HPV vaccines successfully lower the risk of developing cervical cancer by 70% and the risk of developing genital warts by 90%, a condition that can predispose the cervix to a precancerous state [12].

As of 1 January 2021, Cervarix, the HPV-2 vaccine, is refunded in Poland for 138.18 zloty (about 30 EUR) per dose for individuals who are 9 years old and older [13]. Gardasil, the HPV-4 vaccine, and Gardasil-9, the HPV-9 vaccine, are currently not refunded in Poland, though they are available. For the Gardasil-9 vaccine, one requires a doctor referral and needs to pay 340.00 zloty (about 73 EUR) per dose [14] (Figure 3).

#### 3.1.2. Secondary Prevention

The secondary prevention of CC involves screening tests detecting precancerous lesions with subsequent treatment using ablative or excisional methods [15]. Currently, there are three methods of cervical precancer screening: cytology-based screening, molecular HPV screening, and visual inspection with acetic acid. Cytology-based screening is taking a sample from the cervix and placing it either on a slide (Pap smear) or in a container of preservative solution (liquid-based cytology, LBC). If there are cell abnormalities discovered, they are classified by the Bethesda System. Molecular HPV testing requires collecting samples of cells with a small brush, placing them in preservative solution, and processing them in laboratory settings. A visual inspection with acetic acid (VIA) observes cell changes that become visible, mainly because they become faintly white after applying dilute (3–5%) acetic acid with a cotton swab. All of those tests require speculum and light sources.

Since negative results of screening of the CC methods are not always associated with cancer, a further diagnosis of the changed area is needed. There are three diagnostic methods currently used: colposcopy, biopsy, and endocervical curettage (ECC). Colposcopy involves examination of the cervix, vagina, and vulva under strong lights and magnification; a biopsy requires the removal of a sample of previously visible cells changed during VIA, and ECC is scraping of the cells from the endocervical canal, mainly when the transformation zone cannot be observed. The treatment methods for precancerous changes involve cryotherapy, the loop electrosurgical excision procedure (LEEP—removal of the lesion and entire transformation zone), and cold knife conization (CKC—removal of the cone shaped area including ectocervix and endocervix) [16].

In Poland, it is recommended to perform a Pap smear or LBC in women less than 30 years old every 1–3 years and LBC every 1–3 years in women older than 30 and younger than 70 years old. In the older age group, the Co-Test is recommended after 6–12 months in cases where either the LBC or hrHPV test was positive. In cases where both tests are positive or where the detected changes are associated with p16/Ki67, hrHPV 16, 18, non16, and 18, a LSIL colposcopy is recommended and the Co-Test every 1–3 years in the case of negative results of the colposcopy. In cases where AGC-NOS is detected, there is a recommendation for a colposcopy and endometrial biopsy and a Co-Test within 12 months, followed by regular screening after. In women younger than 30 years old, the recommendations are the same, except for the follow-up cytology 12 months after a negative colposcopy result (Figure 4) [17].

#### 3.1.3. Tertiary Prevention

The tertiary prevention has to do with CC patients and their access to, as well as quality and effectiveness of, care. For a tertiary prevention to be successful, the WHO has identified an effective referral system and good compliance with the treatment, as well as functioning palliative care, to be essential. Referral systems, as well as palliative care, are largely dependent on the resources and structure of the healthcare system in each region. Palliative care can be especially demanding of a system, as it requires a high degree of specialized personnel.

Compliance with treatment is both a good predicting factor for good patient outcomes, as well as a complex problem to tackle. Social factors, such as access to treatment and social relievers (housing during treatment, time off work, etc.), along with consistency regarding treatment facilities, were all positively correlated with increased compliance [18,19]. The psychological factors that were associated with increased compliance were the patient’s sense of benefit from the treatment, their sense of disease severity, and their willingness to avoid complications brought on by their condition. Patients who worry about the side effects of treatment or believe that the disease is uncontrollable, however, tend to show lower compliance with treatments. High health literacy and knowledge of the disease and treatments, as well as a positive patient–prescriber relationship, are also factors that increase compliance [19,20].

Therapeutic vaccinations are a controversial yet promising treatment for recurrent HPV-related cancers. Currently, there are several clinical trials investigating the effect of vaccinations on disease progression. The theoretical background suggests that vaccinations can prevent recurrent cancer manifestations by increasing the cell-mediated immunity in an already infected patient, as opposed to preventing the initial infection. The ideal target for this type of therapy is those HPV-infected and those with preinvasive lesions, as progression to cancer proper can take several decades. It is worth noting that this use of HPV vaccinations is not, at the time of writing this article, approved by the US Food and Drug Administration or the European Medicines Agency, although several clinical trials are currently in progress [21].

### 3.2. Analysis of the Current Situation in Poland with Focus on Never Been Screened Persons

Apart from the systemic ideal situation described above, a significant reason for the ongoing high occurrence of CC is the avoidance of regular CC screening. The indicated psychosocial barriers, which prevent patients from participating in prophylaxis, can be classified into three categories: barriers related to facilities/environment, e.g., difficulties in making an appointment, long distance from home to the facility, and problem with transport; barriers related to the personal characteristics of patients, e.g., problems with the organization of time, additional costs, other priorities, lack of awareness of the significance of prophylaxis, and emotional barriers related to the results of the examination itself; and social barriers, e.g., negative experiences with healthcare professionals in the past and lack of support among family and friends [22].

Access to a gynecologist in Poland is actually difficult. According to the NIK report (based on data from the GUS/Central Statistical Office and NFZ/ National Health Fund, NHF) of 2016 [23], there were no gynecology and obstetrics clinics in many rural communes. The highest percentage of communes with this type of clinic was found in the Silesian Voivodeship, and yet, 28.7% were communes without gynecological clinics in the total number of communes, while, in Podlaskie Voivodship, where accessibility was at the lowest level, the percentage of the communes without gynecology and obstetrics clinics was 78.8%. The data shows that the lack of availability of a gynecologist is most common in rural communes, despite the fact that 40% of women and newborns live there. As a result, in the voivodeships with the highest percentage of communes without clinics—Podlaskie and Lubelskie—there are 27,000 patients per one gynecological clinic in the countryside, and some women have to travel up to 50 km to the nearest one. With the simultaneous problem of communication exclusion, which affects up to 13.8 million Poles [24], CC prevention becomes an interdisciplinary problem, and apart from medical issues, an important action to improve the situation of high CC incidence is increasing the availability of healthcare services.

However, the most common reported barriers [22] were those from the category of the personal characteristics of patients. Simple psychosocial interventions focused on these barriers, such as leaflets and automatic messages discussing barriers and coping with them and automatic messages [25,26], have been shown to influence participation in screening. The positive impact of GPs (general practitioners) trained in communication skills, including discussing psychosocial barriers to changing health habits in patients, was also indicated [27].

It has been shown that personal invitations are an ineffective way of increasing participation in the CC prevention program; in Poland, in 2009, only 5.5% responded to personal invitation to an examination. Women living in rural areas and with lower education resigned more often than women living in cities with higher education; therefore, the information campaign should cover the first group in particular [28]. 

Patients also often feel that this type of screening does not apply to them because of: the stability and length of the current relationships, the feeling that the disease affects elders or that, after menopause, these screening are less important than before, no symptoms, and no cases in the family [29]. The above problem is also illustrated by a long case of treating a lowly educated patient who lives in a small town (Appendix A).

## 4. Discussion

Analyzing data from global cancer incidence reports, CC is one of the most frequent cancers amongst the female sex population. Hence, regarding the worldwide statistics, CC is still a matter of concern. Currently, CC prophylaxis consists of three stages: primary prevention, risk factor management, i.e., vaccination against hrHPV; secondary, populational screening for the early-stage CC or precancer; and tertiary, providing timely follow-up visits in order to prevent further complications. The world’s incidence of CC reflects the immense frequency of HPV infection. Unfortunately, vaccination against HPV is not widely administered. The WHO recommends HPV screening as the most effective and simple secondary prophylactic measure [30].

The following facts that are generally known to the authors are the limitations that significantly complicate the analysis of the starting point of the situation in Poland: patients perform tests outside the NHF program; therefore, reporting the % coverage of the population only with NHF data may differ from the actual screening level in Poland; there is a growing interest in other types of screening tests than a Pap smear, i.e., LBC and HPV tests, which are not included in reporting; low social awareness (What is the real coverage in rural areas? What about people emigrating to Poland?); and a reluctance toward preventive examinations occurs in victims of sexual violence—mostly people with different gender orientations or those HIV-infected.

These above-mentioned issues have never been addressed in the Polish literature, and it should be noted that a review that cannot refer to these data must be incomplete. We also realize that our review is not a systematic one; therefore, it is subjected to selection bias. Despite these issues, we tried to make a diagnosis of the current system and propose a change in the way of thinking to be more oriented towards Polish patients. Our diagnosis is that the patient was not medically educated and was in a socioeconomic environment that offered limited access to prophylactic health services for oncological diseases in the population of importance.

Therefore, we see that there is a need to pay attention to the issues not addressed so far in this article on the type of review.

First of all, we are far from countries such as Australia or our close neighbor—Sweden. Sypień and Zielonka et al. estimated Poland’s HPV vaccine coverage to be less than 10%, with a total of only 22,341 individuals having been vaccinated in 2018 [31].

Australia, in 2023, is set to become the world’s first country where CC will be a rare disease defined by less than 6 cases out of 100,000 inhabitants. Moreover, by 2028, there will be less than 4 new cases out of 100,000 inhabitants (2021-35) [32]. On 1 December 2017, Australia transitioned to the renewed National Cervical Screening Program (NCSP), a program that involved primary HPV screening every 5 years for women aged 25–69 years and exit testing for women aged 70–74 years, with partial genotyping for HPV types 16 and 18 and LBC triage for other HPV types [33].

Sweden has implemented a screening program for anyone assigned female at birth between the ages of 23 and 64, wherein patients are called to give regular cell samples. The cases of cervical cancers have halved since the introduction of the screening program in the 1960s. Furthermore, HPV vaccines for 5th grade girls were introduced to the public vaccination schedule in 2010 (but were not actually administered until 2012) and have remained in use since. As of 2019, Swedish boys are also vaccinated in the same manner as part of a plan to eradicate HPV in Sweden within the next 3–5 years. Compliance rates for the screening was 80% of those called on. Sweden also has a high amount of vaccine compliance, with 77.6% of girls and 71.9% of boys born in 2009 in Sweden having been vaccinated with at least two doses of the HPV vaccine in 2022, according to Folkhälsomyndigheten (Swedish Board of Health) [34]. It is, however, worth noting that the initiative towards vaccination is taken not by patients or by the parents of patients but by healthcare providers themselves, meaning that the effort of getting vaccinated against HPV and other severe diseases has decreased. Sweden has also invested a lot of effort in multilingual public education materials, such as 1177 national care advice line/webpages, as well as other official materials being made available in multiple languages. In terms of screening procedures, Sweden has very recently begun offering at-home HPV screening tests in certain regions as of 2020, although it is difficult to say for certain at this time how this will affect compliance with screening.

Among all the recommendations encountered, in the context of groups that avoid screening, one should start with the simplest: teaching teenagers about the route of transmission of HPV infection and the importance of regular visits to the gynecologist.

−Vaccination calendar—HPV vaccination as recommended and free.−Promoting vaccination in schools.−Involvement of the media (including actors and influencers) in promoting HPV vaccination.−Offering vaccinations to mothers who accompany patients in gynecological offices.−SMS notifications about cytology sent.−Mobile cervical cancer screening units (in case of itinerant shops, popular in Podlaskie and Lubelskie voivodships).

When referring to larger groups, more generalizing is necessary to emphasize certain conclusions and solutions for which there is more evidence.

The main and most frequent suggestion is replacing the conventional cytology with the DNA genotyping test for hrHPV types as the basic screening test. This is recommended by leading health agencies and expert groups—including the WHO, the American Society for Clinical Oncology (ASCO), and the American Society for Colposcopy and Cervical Pathology (ASCCP). HPV testing has a relatively high negative predictive value, which means that screening intervals can be safely extended for HPV-negative patients, which seems to be a more favorable solution for women who are screened for CC [35,36,37].

The use of an LBC, which makes it possible to perform several diagnostics tests from one swab, will reduce the number of necessary visits for patient to one in the qualification stage for possible colposcopy with a biopsy, which would be much less oppressive and time-consuming for patients and would improve cooperation with them [38].

Disseminating the idea of “Self-sampling”, a self-collection of vaginal material by the patient using a dedicated brush and subsequent performance of the HPV genotyping tests, would be a good solution for patients who may have difficulties owing to cultural barriers such as exposing private genital areas to an unknown gynecologist, pain experienced during previous examinations, or past history of abuse [39,40].

The creation of a nationwide information system through which patients would receive notifications about upcoming gynecological appointments would reduce the number of patients who irregularly perform CC screening. Additionally, by using such a platform, which could work as an application on a smartphone, patients could receive the results of their latest cytology and collect all previous results in one place [15].

Extending the participation of non-gynecologists (general practitioners and other specialties, midwives, and community nurses) in educating the society about the benefits of secondary CC prevention and promoting the idea of self-collection methods is an effective alternative to material collection by medical personnel.

Additionally, combining CC prevention with breast cancer prevention in a joint program of preventive examinations would be justified in Poland (representative of the Agency for Health Technology Assessment and Tariff System, AOTMiT, personal communication).

On the basis of the patient’s case (Appendix A), we should emphasize the importance of the adequate education of women regarding the possibility of primary and secondary preventions of cervical cancer and how important it is to promote vaccinations among parents, girls, and also boys [41].

In developing preventive programs, education and social activity are very important for socioeconomics, language barriers, and traditional understandings of health and disease. There should be equal access to preventive screening for all women, regardless of race or being foreign-born, uninsured, less educated, and socioeconomically disadvantaged. Important in this regard are free and generally available services. The campaign promoting screening among Haitian women living in South Florida shows that there is a need to disseminate knowledge among women who, for various reasons, do not have access to it. Such campaigns help to develop promising strategies for encouraging the uptake of cervical cancer prevention services [42].

In 2019, the National Comprehensive Cancer Network (NCCN) collaborated with the Maria Sklodowska Curie National Research Institute (MSCNRIO) in Warsaw and Alliance for Innovation (AFI) to create a pilot project called the Polish Edition: The NCCN Guidelines for Cervical Cancer. The project was shepherded by Polish oncologists, including the Health Technology Assessment Agency, National Health Fund, Ministry of Health representatives, and Polish patient advocacy groups. The standards being developed are intended to improve global cancer care so that cancer patients have access to treatments at similar levels in their countries. The Polish edition of the guidelines is one of the first to have been developed by a group of international experts and that will be applicable outside the United States. However, this is a version adapted to Polish conditions (mainly financial) (v. 1.2021) [43]. However, there are no screening guidelines in this document, and the update of the discussion on how to deal with the patient, including, as we understand it, about prophylaxis of all types, from 2019 has not ended at the time of writing this article.

In Poland, every woman between the ages of 25 and 59 who has not had a pap smear screening test in the last 3 years can have this test done in a gynecological office and also in an internal medicine office if it is performed by a midwife certified to perform this test. In the above cases, a referral is not needed [44]. If that information was widely communicated by family doctors, more women would be willing to take the test, and in cases of the results suggesting any pathologies, the treatment could be implemented earlier. Since not every internal medicine office has a midwife or gynecologist hired, and family doctors do not have to give their patients referrals, it may cause situations where patients believe that, if the test is needed, they would receive a referral; doctors don not remind patients to do the tests, however, because it is not part of their specialty, and due to that, the test itself is reserved for gynecologists and midwifes. 

The only refunded HPV vaccine in Poland is Cervarix, and the patient has to cover 50% of the costs. Girls 9–15 are recommended to receive two doses at 5 and 13 months. For women aged 15 and over, three doses at 0, 1, and 6 months are recommended. An internal medicine doctor can qualify the patient for the vaccination. The vaccine is not obligatory [45]. The promotion of the vaccination and the insurance coverage should also be done by the family doctors and, in this case, the pediatricians.

The implementation of screening based on HPV genotyping in Poland requires a huge organizational effort. This test is more sensitive but less specific compared to cytology, which may increase the risk of false-positive results and, thus, overtreatment, which is particularly undesirable in women with maternity plans. In order to prevent this, the national CC prevention program should be carefully and gradually modernized, which should be preceded by pilot studies with their appropriate analyses. The necessity of pilotage is mentioned in *Supplements to European Guidelines for Quality Assurance in Cervical Cancer Screening*, published in 2015. Many organizational, logistic, economic, clinical factors, and considerations should be considered before the introduction of HPV-based screening. In Poland, the recruitment of patients for the HPV-DNA test pilot study is currently underway, organized by the Central Coordination Center, Maria Sklodowska Curie National Research Institute as part of the National Oncological Strategy under an agreement with the Minister of Health [46].

Ultimately, screening tasks should be elaborated and carefully counted. This is beyond the scope of this article. However, we must point out that the expert and advisory teams still focus on medical issues, disregarding psychocultural considerations at least to the extent that we indicated in this review (Figure 5).

## 5. Conclusions

The current administration of CC screening and prevention is deemed inadequate. With the state implementing the improvements discussed earlier, a larger audience could potentially be reached and be provided with the necessary incentives for partaking in CC screenings. Along with such measures, better networking amongst cervical screening clinicals may contribute to the desired applicable results. In closing, such benefits can lead to the reduction of costs in tertiary prevention.

## Figures and Tables

**Figure 1 diagnostics-12-01807-f001:**
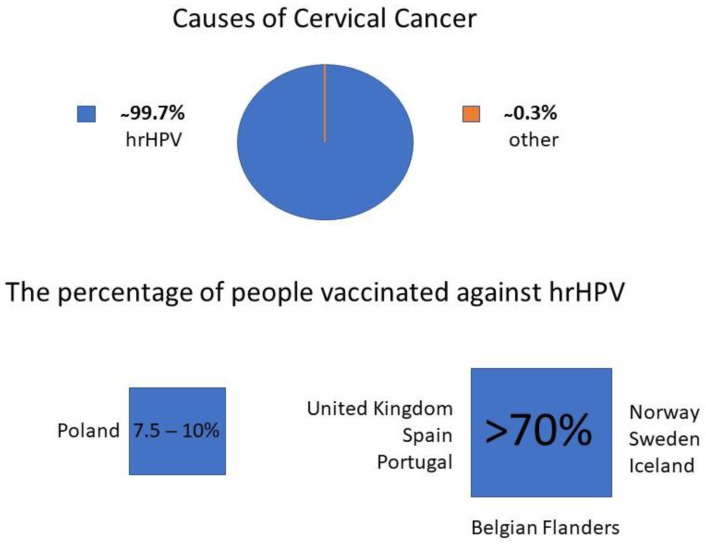
The level of the lack of protection of the Polish population against the main risk factor for cervical cancer.

**Figure 2 diagnostics-12-01807-f002:**
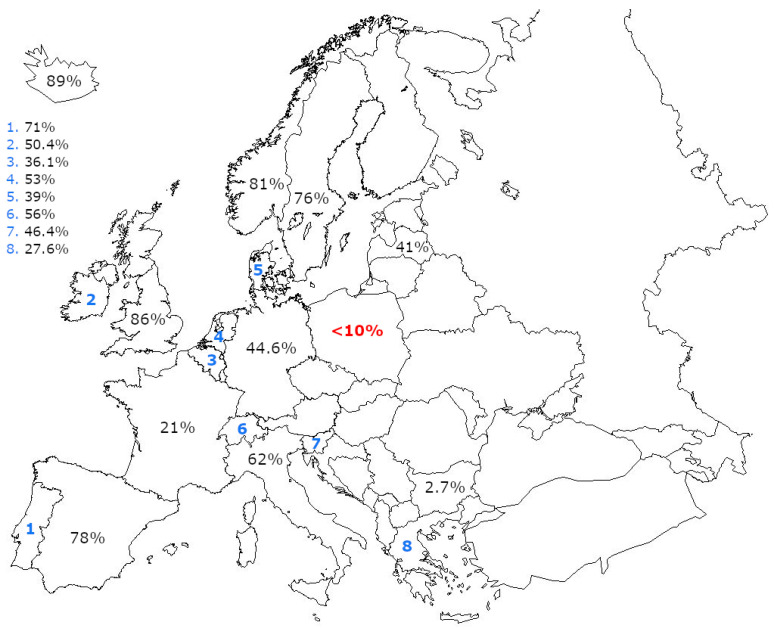
A map of the ratio of vaccination among populations of EU countries.

**Figure 3 diagnostics-12-01807-f003:**
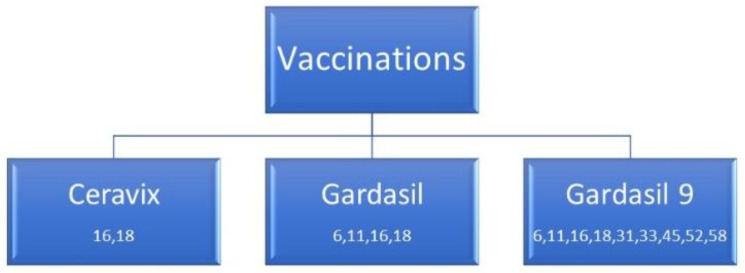
An overview of the available vaccines in Poland.

**Figure 4 diagnostics-12-01807-f004:**
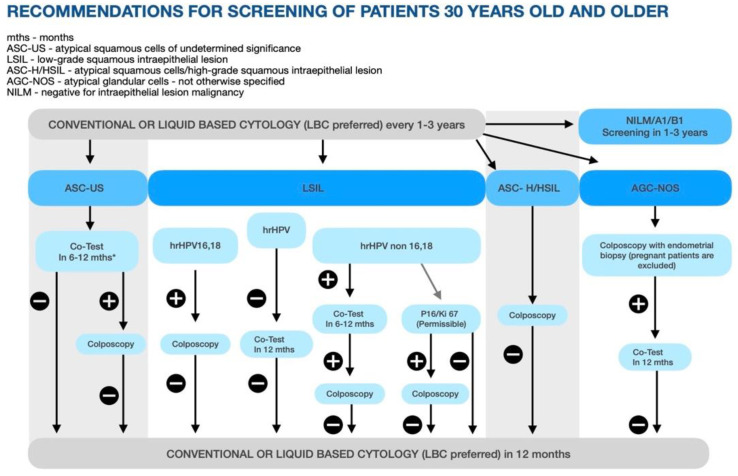
Recommendations for the screening of patients 30 years old and older in Poland.

**Figure 5 diagnostics-12-01807-f005:**
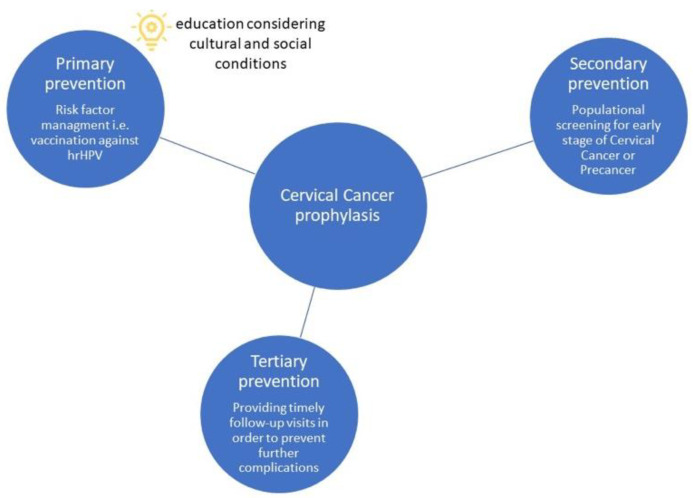
Review of the types of prevention with an appeal to introduce changes in the communication channel with the exposed part of the Polish population.

## Data Availability

Not applicable.

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
