# Peer review of "Thousands of Women’s Lives Depend on the Improvement of Poland’s Cervical Cancer Screening and Prevention Education as Well as Better Networking Strategies Amongst Cervical Cancer Facilities"

_diagnostics, 2022, doi:10.3390/diagnostics12081807_

Round 1
Reviewer 1 Report
The article, although of relatively local interest, presents very interesting and important data regarding cervical cancer in Poland. On account of the fact that Poland is a country that combines a large economic development during the last decades together with a stably large proportion of rural population, it presents a very good model to examine how people prevent cervical cancer. Thus the study deserves publication. I have some comments and suggestions that I recommend to be addressed and further discussed:
The authors state that WHO recommends HPV screening as the most effective prophylactic measure. However, according to the presented data, it seems to me that promotion and refunding of vaccines as well as access to the gynecologist are probably among the factors that if resolved would decrease the incidence of cervical cancer. I suggest to discuss this fact in an effort to present the importance of the country as a model case to examine how the public system reacts and responses to the incidence of the disease
A few minor comments
line 99 and elsewhere: Since the authors present data regarding EU, maybe it would be better to replace “Europe” with “EU”
line 100: please replace “our country” with “Poland” or another term
line 121 as well as in the abstract: please define and explain better the term “white papers”
Author Response
Please find enclosed reviewer's notes and our reply included in this document.

Reviewer 2 Report
This review paper entitled “Thousands of Women’s Lives Depend on The Improvement of Poland’s Cervical Cancer Screening and Prevention Education as Well as Better Networking Strategies Amongst Cervical Cancer Facilities” is well organized and comprehensively described. I suggest accept in present form.
Author Response
Please find enclosed document.
